# Modular Chimeric Antigen Receptor Systems for Universal CAR T Cell Retargeting

**DOI:** 10.3390/ijms21197222

**Published:** 2020-09-30

**Authors:** Ashley R. Sutherland, Madeline N. Owens, C. Ronald Geyer

**Affiliations:** 1Department of Biochemistry, Microbiology and Immunology, University of Saskatchewan, Saskatoon, SK S7N 5E5, Canada; ashley.sutherland@usask.ca (A.R.S.); mno167@usask.ca (M.N.O.); 2Department of Pathology and Laboratory Medicine, University of Saskatchewan, Saskatoon, SK S7N 5E5, Canada

**Keywords:** chimeric antigen receptor (CAR T), universal CAR T, modular CAR T, universal immune receptor, CAR adaptor, adoptive immunotherapy, antibody, split CAR

## Abstract

The engineering of T cells through expression of chimeric antigen receptors (CARs) against tumor-associated antigens (TAAs) has shown significant potential for use as an anti-cancer therapeutic. The development of strategies for flexible and modular CAR T systems is accelerating, allowing for multiple antigen targeting, precise programming, and adaptable solutions in the field of cellular immunotherapy. Moving beyond the fixed antigen specificity of traditional CAR T systems, the modular CAR T technology splits the T cell signaling domains and the targeting elements through use of a switch molecule. The activity of CAR T cells depends on the presence of the switch, offering dose-titratable response and precise control over CAR T cells. In this review, we summarize developments in universal or modular CAR T strategies that expand on current CAR T systems and open the door for more customizable T cell activity.

## 1. Introduction

Engineering T cells to express chimeric antigen receptors (CARs) has shown wide-ranging potential as a potent anti-cancer therapeutic. Characteristically, CARs consist of an extracellular antigen-binding single-chain antibody variable fragment (scFv) and hinge region linked to transmembrane and intracellular signaling regions. This engineered construct fuses the specificity of an antibody to T cell-effector functions, allowing for target cell lysis, release of cytokines, and T cell proliferation [1,2]. In clinical trials, CAR T cell therapy has shown remarkable success in treating hematological malignancies by targeting B cell antigen CD19. Numerous studies showed high remission rates, rapid tumor eradication, and durable responses in patients with refractory disease, raising expectations for expanding the types of cancers that can be treated with CAR therapy [3,4,5]. Although these results are encouraging, several challenges exist that inhibit the broad application of this treatment. Firstly, tumor heterogeneity is a complication that hinders CAR T development. Conventional CAR T cells have a fixed, single-antigen targeting ability, making the therapy vulnerable to antigen-loss relapse due to downregulation or antigen deletion [6,7]. CAR T therapy targeting a single antigen may initially demonstrate tumor regression; however, many cases have been reported in clinical trials of antigen-negative relapse after CD19 CAR T therapy due to tumor antigen escape [8,9]. Engineering of CAR T cells against a variety of tumor-associated antigens (TAAs) is a method to overcome tumor immunoediting, yet this approach comes with its own set of challenges. Further clinical success of CARs would necessitate the engineering of T cells tailored for each patient targeting various TAAs. However, significant technical requirements and financial costs involved in the generation and optimization of CARs directed at individual antigens limit this approach’s usefulness. To circumvent the technical and economic challenges of individually manufacturing and testing each new CAR, creating a platform using ‘universal’ redirected T cells against virtually any cell surface antigen is of particular importance for the rapid screening in pre-clinical models and the broad application of CAR T therapy.

The ability to generate modular or universal CARs hinges on the separation of targeting and signaling elements. Modular CAR T cells are not targeted at the tumor antigen itself; instead, the CAR is directed at an adaptor or switch element (Figure 1). This adaptor serves as the targeting element, binding to the tumor antigen, and is required to bridge the immunological synapse. The firing of the CAR T should occur only in the presence of the switch, and swapping out the adaptor molecule allows for redirection of the T cell without the need for re-engineering and time-consuming remanufacturing. This modular treatment approach offers the possibility for flexibility in tumor targeting in the clinic; adapting with the patient’s tumor by adjusting treatment based on the cancer’s changing antigen expression could be envisioned. Fixed antigen targeting often hinders cancer treatment, with this modular CAR T approach driving an already innovative biological therapy into truly tailored cancer therapy.

In the toolkit of building modular CARs, the number of adaptors has been increasing, with components such as immunoglobulin (IgG)-based adaptors such as scFvs (single-chain variable fragment), Fabs (antigen-binding fragment), nanobodies, and full-length IgGs being the most established. Both antibody-based and targeting ligand adaptors have been redirected using tags attached either genetically or post-translationally and include peptide tags such as neo-epitopes, SpyTag, leucine zippers, biotin and fluorescein isothiocyanate (FITC). Repurposing clinically approved IgGs and adding redirecting tags could reduce the regulatory hurdles and allow for a suite of targeting elements that clinicians could employ for a wide variety of cancer indications.

Improving the safety profile of CAR T therapy is a rapidly developing area of research with the goal of expanding the therapy to treat a broader range of cancer patients. With the promise of CAR T therapy comes life-threatening side effects, including severe cytokine release syndrome (CRS), neurological toxicities and organ failure, resulting from the unrestrained proliferation of CAR T cells [10,11]. Toxicities related to on-target off-tumor reactions can occur when low levels of the targeted cancer antigen found on normal tissue are targeted by the engineered T cells. When toxicities become severe enough, administration of high-dose corticosteroids is required, decreasing the T cell numbers [10,11,12]. Research in the area of ‘next-generation’ CAR T cells has incorporated methods such as suicide genes or switches as a means to eliminate T cells. These ‘emergency stop’ methods may avert a lethal outcome to the patient; however, the drawback is that the CAR T cells are eliminated, and with it any therapeutic response—a bad ending for a costly and time-consuming therapy. Universal or modular CAR T strategies that require administration of an adaptor may fit the criteria for better control of engineered T cells. The ability to titrate on adaptors could facilitate the ‘turning off’ of CAR T cells by halting administration of the adaptor, possibly enhancing their safety profile without the need to destroy the T cells. Additionally, this tunable response could better manage side effects as well as fine tuning of CAR T activity. These approaches for CAR T generation allow for conjugation of the tagged targeting element with the anti-tag CAR T. This enables targeting multiple TAAs and flexibility in administration of the targeting element, a step towards overcoming current clinical limitations of CAR T therapy.

In this review, we summarize emerging systems developed to overcome the limitations of CARs with fixed antigen specificity using universal CAR T strategies. This modular approach using adaptors to ‘turn on’ CARs enables targeting multiple TAAs with one receptor, imparting near-limitless antigen specificity and fine tuning of CAR activity.

## 2. Modular CAR T Platforms

### 2.1. Biotin-Binding Immune Receptors

Exploiting one of nature’s strongest non-covalent bonds, the avidin–biotin interaction has been used to generate a universal tumor-targeting system, biotin-binding immunoreceptor (BBIR) (Figure 2A). BBIRs constructed by Urbanska et al. [13] consist of an extracellular modified dimeric avidin (dcAv) linked to an intracellular T cell signaling domain. This split design allows for easy target modification and targeting of multiple antigens. Monomeric avidin was unable to elicit an effector response, most likely due to the decreased affinity between biotin and monomeric avidin.

Tumor cells in vitro are either pre-targeted or co-administered with biotinylated antigen-targeting molecules (IgG antibodies, scFvs, or other tumor-specific ligands), and T cells expressing BBIRs bound specifically to exert effector cell functions. In vitro functioning of dcAv BBIR is comparable to traditional CAR T for cell lysis and cytokine secretion, illustrating this system’s utility to target cells and perform effector functions. Significantly, BBIR cells generated a dose-dependent response with the addition of adaptors. When tested against a panel of cells with variable TAAs, BBIRs could target antigens both simultaneously or sequentially, showing the tunability of CAR response and their utility in antigen escape scenarios. Furthermore, the authors proposed a platform using BBIRs to screen candidate antibodies or other targeting elements in vitro for rapid pre-clinical screening. BBIRs performed similarly in vivo in a xenograft mouse model of human ovarian cancer. As CAR safety is essential for moving into the clinic, BBIRs were exposed to supraphysiological levels of biotin and showed no antigen-independent activation. Interestingly, T cells could not be ‘pre-armed’ with the biotinylated targeting element; the adaptor had to be either pre-targeted to coat the tumor surface or co-administered with BBIR T cells.

Lohmueller et al. [14] drew on this system and further affinity enhanced streptavidin, designing a biotin-binding domain where monomeric streptavidin could be used in the CAR system with higher affinity to biotin than the dimeric form. This affinity-enhanced form, mSA2, has a more compact structure than the dimeric form employed previously, possibly increasing the type of antigens able to be targeted by this system. The mSA2 CARs were able to distinguish antigen-positive cells precoated with biotinylated antibody in vitro and produce a specific effector response. Antigen-negative tumor cells did not elicit an effector response, neither did non-binding biotinylated antibody, showing that in order for CAR T cells to be turned on, the antibody must be bound to the target cells. The mSA2 CAR T cells showed potent effector functions; however, its potential immunogenicity could hamper its adoption in the clinic. With the BBIR system, excess biotin does not impart an inhibitory effect, and therefore could not be used as a potential ‘off switch’ for added safety.

### 2.2. Anti-FITC CAR Strategy

Fluorescein isothiocyanate (FITC), derived from fluorescein, is a fluorescent label commonly used to tag antibodies. Tamada et al. [15] first exploited this common labeling method and generated anti-FITC CAR T cells able to be directed by FITC-tagged antibodies (Figure 2B). The extracellular portion of the CAR is comprised of an anti-FITC scFv that recognizes FITC-labeled cetuximab (anti-EGFR), trastuzumab (anti-HER2), and rituximab (anti-CD20), antibodies that are already employed clinically. Redirected CAR T cells were found to be effective both in vitro and in vivo to specifically bind their respective tumor cells and exert anti-tumor effects. This system showed effectiveness at targeting multiple TAAs to better address heterogenous cancer populations. Additionally, the use of anti-FITC CAR T cells was shown to restore the usefulness of monoclonal antibodies to additional cancer types. They discuss that in patients with *Kras* mutations, cetuximab does not provide therapeutic benefits; however, when cetuximab is utilized with anti-FITC CAR T cells, anti-tumor effects are shown, as illustrated with the SW480 cell line that containing a *Kras* mutation. The anti-FITC CAR T system was applied using trastuzumab—Cao et al. conjugated FITC to trastuzumab in a site-specific manner compared to another strategy where a peptide neo-epitope (PNE) was fused to trastuzumab [16]. Both antibody tagging methods showed a dose-titratable immune response, capable of completely clearing HER2-positive tumors in vivo. The first clinical use of trastuzumab incorporated into a CAR T resulted in a serious adverse event, with the patient developing on-target, off-tumor toxicity related to the redirection of CAR T cells to lung epithelium, proving fatal [17]. Since this initial trial, many groups have investigated safer ways to target HER2, reviewed by Liu et al. [18], with the modular anti-FITC CAR T technology, a contender to address the safety issues with targeting this cancer-associated antigen.

Expanding the targeting elements to more than full-length antibodies, Zhang et al. employed switchable CAR-engineered T cells using anti-tumor peptides that specifically target integrin avβ3 through an 18-amino acid sequence fused to FITC [19]. This peptide adaptor molecule, termed FITC-HM-3, specifically targeted tumor cells and regulated CAR T cell activity. Demonstrating that low-molecular-weight switch molecules can be effective at redirecting engineered T cells, Lee et al. [20] employed a cocktail of small bifunctional molecules in conjunction with anti-fluorescein CAR T cells to target cancer cells in vitro and in vivo. The bifunctional molecules, called CAR T cell adapter molecules (CAMs), consist of fluorescein linked to a tumor-specific ligand through a hydrophilic spacer. The use of a mixture of CAMs enables the targeting of heterogenous solid tumors and broadens the applicability of CAR T cell therapy by using small molecules, which could improve tumor penetration, as opposed to larger full-length antibodies. Additionally, improved safety is offered by the short half-life (~90 min) of small molecules, allowing them to rapidly clear from receptor-negative tissue.

Optimizing the complex between the CAR T cell, switch, and tumor antigen is essential for optimal CAR T activation and cell killing. Using the modular CAR system, Ma et al. [21] utilized anti-FITC CARs to target both CD19 and CD22, whereby antibody fragments were site-specifically modified with FITC through genetically encoded non-canonical amino acids. This allowed for the incorporation of FITC to optimize of the geometry of the immunological synapse. Compared head to head, the optimized anti-FITC CAR T targeting CD19 performed similarly to conventional CD19-targeting CAR T, necessary for moving this technology forward into the clinic. Furthermore, excess FITC at 10 μM was shown to dampen CAR T activity in vitro, a feature that could be used to improve safety in the clinic. Others have shown that the addition of FITC-labeled non-specific antibodies could also be used to attenuate CAR T cells [15].

The targeting of folate receptors using anti-FITC CARs has been demonstrated by several groups [22,23,24]. Lu et al. [23], using FITC conjugated to folic acid as the switch molecule, modeled severe cytokine release syndrome and determined that CRS could be alleviated through the titration of the folate FITC adaptor or by intermittent dosing. Reversal of severe CRS could be achieved by intravenous sodium fluorescein to transiently interrupt CARs, without destroying the engineered T cells. With the ability to shut down the CAR T response through the addition of FITC [21], FITC labeled non-specific antibodies [15] or sodium fluorescein [23,24], this system with its added ‘safety switches’ could allow for engineered immune cell deactivation if toxicity develops, possibly being able to salvage the therapy by re-administering the switch molecules. While encouraging, the possible immunogenicity of FITC adaptors in the context of CAR T systems requires further study.

### 2.3. The SpyTag-SpyCatcher Universal CAR T System

The SpyTag/SpyCatcher protein ligation system employs a unique peptide: protein ligation reaction to link the tagged targeting element to the immune receptor. In 2012, Zackeri et al. reported a fibronectin-binding protein from *Streptococcus pyogenes* that, upon splitting it into two parts, followed by rational engineering, an N-terminal protein fragment (SpyCatcher) and a C-terminal 13-amino acid peptide (SpyTag) were produced [25]. The two parts will spontaneously reconstitute to form an isopeptide bond without the need for co-factors, enzymes, or specific conditions. Minutolo et al. exploited this system to generate a SpyCatcher immune receptor [26] (Figure 2C). This immune receptor contains the SpyCatcher protein as the extracellular domain, linked to intracellular signaling regions. TAA-specific targeting ligands, such as IgG antibodies, are site-specifically labeled with SpyTag. This post-translational covalent assembly allows for the redirection of T cells to multiple TAAs to exert targeted effector cell functions. The SpyCatcher immune receptor activity depends on the presence of both target antigen and SpyTag-labeled targeting element, allowing for titratable control of the engineered T cells. Arming the SpyCatcher CAR T cells with SpyTagged antibodies showed receptor levels decreasing over time, with complete loss observed after 96 h. SpyCatcher CAR T cells were shown to become functional upon the addition of SpyTag targeting element and lyse antigen-expressing target cells in vitro. Using an immunodeficient mouse xenograft model, the authors showed that HER2-positive xenografts could be targeted with SpyCatcher CAR T cells pre-armed with SpyTagged Herceptin. Additional targeting ligand was administered every three days, and administration throughout treatment was shown to be necessary for tumor clearance. The SpyTag/SpyCatcher system was tested using targeting elements against HER2, EGFR, EpCAM and CD20 and Liu et al. expanded the range of targetable antigens by demonstrating SpyCatcher immune receptors could be constructed to target the hepatocellular carcinoma antigen, human glypican-3 (hGPC3) using a SpyTagged anti-hGPC3 scFv [27].

Potential immunogenicity is an issue that may hamper SpyCatcher immune receptor adoption in the clinic. Owing to its bacterial origin, the Tag/Catcher system may be vulnerable to recognition by the patient’s immune system. Work has been performed in developing SpyCatcher/SpyTag variants with truncations aimed at reducing potential immunogenicity [28] and tested using immunocompetent mice, but further study is needed to determine the likelihood of adverse reactions in humans.

### 2.4. Leucine Zippers to Retarget CAR T

Cho and colleagues developed a split, universal, and programmable (SUPRA) CAR system that allows for a modular platform to tune the specificity and activation of CAR T cells [29] (Figure 2D). The system consists of two parts: (1) a universal receptor (zipCAR) expressed on T cell surfaces (2) and a tumor-targeting scFv adaptor (zipFv). The zipCAR component consists of intracellular signaling domains linked through a transmembrane domain to an extracellular leucine zipper. The zipFv is a fusion protein consisting of an scFv and a leucine zipper that can interact with the leucine zipper of the zipCAR.

The split CAR design has several variables that can be manipulated to modify the specificity and adjust the SUPRA CAR T cells’ activity. The leucine zipper pairs’ affinity correlated with cellular activation as determined by cytokine secretion and target cell killing efficiency. Cells with higher zipCAR expression showed greater cytokine secretion upon activation. The authors showed that SUPRA CAR T cell activity could be inhibited using competitive zipFvs with leucine zipper domains that bind the original zipFv but not the zipCAR, thus preventing the activation of zipCARs. The activation level of SUPRA CAR T cells could be further modified by changing the competitive zipFv’s leucine zipper’s affinity for the original zipFv. SUPRA CAR T cells can also be used to increase tumor specificity through combinatorial antigen sensing, wherein more than one zipFv is introduced. Different signaling domains could be controlled by orthogonal SUPRA CARs, where zipCARs consisting of different intracellular signaling domains and leucine zipper components to specifically activate certain pathways upon antigen binding. The in vitro and in vivo effectiveness of the zipCAR platform was demonstrated; however, further studies, including clinical trials, are needed to confirm its efficacy and safety.

### 2.5. ConvertibleCAR Strategy Using Modified NKG2D Extracellular Domain

This modular CAR T cell variant, termed *convertible*CAR T cells, uses an inert form of the NKG2D extracellular domain as the ectodomain of the CAR [30] (Figure 2E). NKG2D, an activating receptor expressed on NK cells and some myeloid and T cells, was mutated in its ectodomain such that it cannot engage naturally occurring ligands. This mutant is deemed inert and referred to as inert NKG2D (iNKG2D). Orthogonal ligands were selected that specifically engaged iNKG2D but not WT NKG2D. Antibodies were fused to the orthogonal ligand, U2S3, to generate bispecific MicAbodies, which can specifically direct and activate iNKG2D-CAR-expressing T cells upon binding the respective antigen on a target cell surface. This modular system allows *convertible*CAR T cells to be redirected to different antigen-positive target cells. Additionally, the U2S3 ligand can be fused to payloads, such as cytokines to be delivered to iNKG2D-CAR-expressing cells, promoting their expansion. Expanding the repertoire of modular CAR T systems to more than cancer, Herzig et al. demonstrated that a *convertible*CAR T approach can be utilized to effectively and specifically kill HIV-infected CD4 cells [31]. Despite the advances in antiretroviral therapy, the primary obstacle to curing HIV-positive individuals is latently infected cells that persist. In an effort to target this reservoir of HIV-infected cells, the authors employed *convertible*CAR T cells armed with anti-HIV antibodies fused to an orthogonal MIC ligand. This allowed for the specific killing of HIV-infected primary CD4 T cells in vitro, only when the *convertible*CAR T cells and MicAbodies were bound. The use of a modified human activating receptor, NKGD2, may reduce the chance of an immunogenic reaction in the clinic, but *convertible*CAR system’s safety has yet to be established.

### 2.6. SNAP CAR Strategy: Enzymatic Self-Labeling CARs

As one of the newest methods to be utilized for the generation of modular CAR Ts, an enzymatic strategy to link adaptor to CAR signaling regions is employed, where the CAR’s extracellular domain contains the self-labeling SNAPtag enzyme [32] (Figure 2F). Antibodies are conjugated with benzylguanine (BG), with which the SNAP enzyme can react and form a covalent bond. SNAPtag is a modified human O-6-methylguanine-DNA methyltransferase, engineered to react with BG. Potent effector cell activity was shown when SNAP-CAR T cells were co-cultured with antigen-positive target cells, both with BG-conjugated full-length IgG antibodies and Fab fragment. An advantage to this anti-tag CAR system is the formation of a covalent bond between the SNAP enzyme and the BG moiety on the antibody.

Along with the SpyTag/SpyCatcher system, SNAP CARs are distinct from the other anti-tag systems, which rely on a transient interaction between antibody-tag conjugate and the CAR-modified cells. Additionally, the SNAP protein is of human origin and thus is unlikely to be immunogenic. A similar system was designed using the synthetic Notch receptor (synNotch) instead of the T cell receptor, wherein upon antigen binding, the Notch core protein is cleaved by endogenous proteases and releases a transcription factor from the cell membrane where it then travels to the nucleus to carry out its transcriptional regulation function. The generated self-labeling synNotch receptor, containing the SNAPtag protein covalently fused to the adaptor antibody, similar to the SNAP CAR system, functions as modular platforms for switchable CAR T activity.

### 2.7. The Co-Localization-Dependent Protein Switch (Co-LOCKR) CAR T System

Utilizing a novel logic-gated system, Lajoie et al. designed a protein switch system termed co-localization-dependent protein switch (Co-LOCKR), whereby the switch is engaged through a conformational change only when conditions are met, allowing for the implementation of AND, OR and NOT logic gates for precision target cell killing [33] (Figure 2G). The switch consists of a ‘cage’ protein, which sequesters a functional ‘latch’ peptide in an inactive conformation. The binding of a separate ‘key’ protein will induce a conformational change such that an effector protein or CAR T cell can interact. The authors employed a Bim-Bcl-2 interacting pair where the CAR contains a Bcl-2 binder which interacts with the Bim contained on the latch peptide. Using designed ankyrin repeat protein (DARPin) domains to target the cage element to HER2 and two key domains to EGFR and EpCAM, only cells that co-express both antigens (either HER2 + EGFR or HER2 + EpCAM) will activate Co-LOCKR. The cage domain containing the latch with Bim peptide will only be exposed once the key element interacts, initiating Bcl-2 CAR binding and subsequent T cell activation. Using target cells that express combinations of HER2, EGFR and EpCAM in a mixed population, a HER2-EpCAM Co-LOCKR showed that it would preferentially kill cells expressing both HER2 and EpCAM, and not those expressing HER2 and EGFR, or only HER2 or only EpCAM. The same experiments were performed with HER2-EGFR Co-LOCKR showing target cell killing by Bcl-2 CAR was restricted to only cells containing both HER2 and EGFR. These experiments demonstrated that CAR T cells were engaged specifically when in the presence of target cells co-expressing the correct pair of antigens and the degree of CAR expansion correlated with the density of antigen. The authors showed that between 2.5 nM and 20 nM of Co-LOCKR could be used without causing off-target CAR T cell killing, and the sensitivity of the switch could be further tuned through altering the cage-latch and cage-key affinities.

This method has the ability to reduce off-target effects and precisely direct CAR T activity towards specific target antigens. Additionally, decoy proteins fused to a targeting domain against an antigen to be avoided could be created, allowing for a CAR T ‘off switch’, where NOT logic is employed as the decoy sequesters the key, preventing cage activation and CAR T firing. In vivo experimentation is needed to evaluate the system’s efficacy, and further studies on the potential immunogenicity of the designed proteins are required to broaden the application of this modular CAR T system.

### 2.8. Anti-5B9 Tag CAR: The UniCAR Platform

Using a unique peptide derived from an autoantigen to redirect CAR T cells, a novel modular CAR system, termed UniCAR, was developed [34] (Figure 2H). The system consists of two components: (1) a CAR with an anti-La protein scFv and (2) specific targeting modules (TMs) that redirect UniCAR T cells to targeted tumor cells. The anti-La protein recognizes a short non-immunogenic peptide motif of ten aa (5B9 tag) derived from the human nuclear autoantigen La/SS-B. TMs consist of a binding domain, such as a tumor-specific scFv, fused to the 5B9 tag that is recognized by the scFv portion of the UniCAR. UniCAR T cells are inactive when TMs are absent, providing an ‘on’ and ‘off switch’ to effector cell activity. The variety of antigens targeted by this system illustrates the flexibility of this approach; CD33, CD98, CD123, FLT3, EGFR, STn, GD2, PSMA, and PSCA [34,35,36,37,38,39,40,41,42,43] have all been targeted with the UniCAR platform. An advantage of the UniCAR system is the inherent safety switch by halting the infusion of TMs, and the flexibility that can enable targeting of tumor escape variants by using bispecific targeting agents.

Cartellieri et al. used the UniCAR system to target acute myeloid leukemia (AML), a heterogenous leukemic disease [34]. A previous analysis showed that nearly all AML blasts are positive for CD33, CD123, or both, and scFv-based TMs were designed to target these antigens. Additionally, a dual-specific TM was engineered to target CD33 and CD123 and was shown in a cytotoxicity assay to lyse AML cell lines more effectively than equal molar ratios of each monospecific TM. Using this bispecific strategy may help reduce the risk of tumor escape variants. In the absence of TMs, the UniCAR T cells remained inert in vivo and did not show signs of toxicity.

As a more ‘off-the-shelf’ therapeutic approach, Mitwasi et al. demonstrated that UniCAR platform could be adapted to the NK-92 cell line [44]. The TM targeted the disialoganglioside GD2. Two types of TMs were explored: an scFv form as previously described, and a novel homodimeric format wherein the E5B9 epitope is connected to the GD2-specific antibody domain via an IgG4 Fc region. This novel TM format was used due to its longer half-life compared to the scFv version; the longer half-life was desired to be closer to that of the NK-92 lifespan due to their limited in vivo persistence. Therefore, they do not require a fast safety switch and the longer half-life of the adaptor reduces the need for continuous infusion. Both versions of the TM led to efficient and specific cell lysis of GD2+ neuroblastoma cells in vitro and in vivo. The use of the NK-92 cell line provides advantages such as lower side effect risk due to their restricted lifespan. Although the 5B9 peptide is derived from an autoantigen, studies completed examining the immune response against the La autoantigen demonstrated anti-La antibodies were not developed and an immune response was not mounted against the 5B9 peptide, as reviewed by Bachmann [45], which would indicate that UniCAR TMs are likely not immunogenic.

Albert et al. examined whether other antibody derivatives could be employed in the UniCAR platform [37]. They incorporated a nanobody targeting EGFR as the targeting module, which effectively retargeted UniCAR T cells to EGFR+ tumor cells and mediated specific cell lysis in vitro and in vivo. The anti-EGFR nanobody was subsequently radiolabeled with ^64^Cu and ^68^Ga, and biodistribution, clearance and stability of the targeting module-UniCAR complex were measured. The experiments established that TMs could be released from the UniCAR and dissociated both in vitro and in vivo in a dose-dependent manner, demonstrating its ability to act as a self-limiting switch. The rapid elimination of the nanobody-based TM could improve its safety profile. This work, along with Loureiro et al. [38] and Ardnt et al. [42], demonstrated the potential for UniCARs to be used for targeted immunotherapy and simultaneously as a PET imaging tool to track CAR T therapy. The PET tracer PSMA-11, which binds prostate-specific membrane antigen, was converted into a UniCAR-TM having a dual function as a CAR retargeting element as well as a non-invasive PET imaging reagent, making this a member of a new class of theranostics.

### 2.9. Anti-PNE CARs: Redirecting T Cells Using Peptide Neo-Epitope Tagging

Similar to the UniCAR platform, anti-PNE CARs take advantage of a peptide tag to redirect CAR T cells. Rodgers et al. first engineered an antibody-based bifunctional switch to be compatible with anti-peptide neo-epitope (PNE) CAR T cells, called switchable CAR-T cells (sCAR T) [46] (Figure 2I). The switch molecule is a 14 amino acid peptide neo-epitope derived from a yeast transcription factor that was shown to have a low probability of inducing an antibody response. The sCAR T cells form an immunological synapse through a PNE engrafted Fab to specifically direct T cell activity to the targeted cells using an anti-PNE scFv CAR. The authors created sCAR T cells directed against CD19 and CD20 and used the system to target B cell malignancies. The switch molecules were systematically optimized, focusing on spatial interactions, to achieve the most efficacious effector/target cell interaction. The authors show that sCAR-T cell activity is titratable, dose-dependent, and strictly dependent on the presence of the switch molecule; all of these characteristics contribute to improved safety compared to that of conventional CAR T cell therapy. Additionally, the codelivery of anti-CD19 and -CD20 switch molecules may prevent antigen escape to more effectively eliminate heterogenous B cell cancers. Viaud and colleagues further studied CD19-directed switchable CARs by examining the factors determining the induction of memory and expansion of sCAR T cells [47]. The formation of a memory population is important to consider, as a naïve, persistent central memory phenotype has been correlated with prolonged remissions in acute lymphoblastic leukemia patients. Conventional CAR T cell therapy cannot achieve the “rest” period required to stimulate a memory phenotype as they are constitutively “on” and interact with antigen. sCAR T cells targeted against CD19+ B cell lymphomas in a competent murine host showed that the timing and dosage of the anti-CD19 switch molecule could promote the expansion and contraction as well as the phenotype of the sCAR T cell population. It was established that a “rest” period used in conjunction with cyclical dosing of switch molecules could induce the production of a memory population, showing potential for enhancing the efficiency and persistence of CAR engineered T cells.

In addition to hematological cancers, anti-PNE CARs have been developed to target solid tumors expressing HER2 [16,48]. Pancreatic ductal adenocarcinoma (PDAC) was targeted with switchable CAR T cells using an anti-HER2 Fab-based switch molecule engrafted with a PNE tag. Using patient-derived xenografts obtained from patients with advanced stage, difficult-to-treat pancreatic tumors, durable remission was achieved by a single injection of switchable CAR T cells and five doses of the switch molecules. Switch molecules were further administered for 10–14 injections, resulting in long-term remission for all animals involved. Switchable CAR T cells persisted even after switch injections were halted, demonstrating switchable CAR T cells have the potential to be effective in safely treating aggressive and disseminated disease. The ability to modulate the dosage of the switchable CAR T may allow for safer targeting of HER2 and other antigens in the clinic.

## 3. Conclusions and Future Perspectives

CAR T cell therapy has already proven itself in the clinic as a powerful anti-cancer therapeutic. Needed to further its expansion into a wider array of cancers types is both identification of new targets combined with innovative CAR T design. For CAR T therapy to realize its potential in solid tumors, addressing tumor heterogenicity is paramount to its adoption in the clinic, along with a means to better modulate CAR activity to enhance its safety profile. An adaptable system such as modular CAR Ts conceivably could address these issues by tailoring CARs to a patient’s specific cancer and adapting treatment using a toolkit of adaptor targeting elements. Simultaneous or sequential targeting of multiple TAAs through universal CARs could mitigate antigen escape, all without the time-consuming and expensive re-engineering of T cells.

Essential for the success of any CAR T approach is the ability to mitigate side effects. Modular CAR activity can be dialed in for enhanced control of engineered T cells and interruption of switch molecules administration, or in some cases using titratable ‘off switches’, able to dampen side effects associated with T cell proliferation. The use of adaptors to redirect CAR T cells expands on the repertoire of possible targeting elements such as using full-length IgGs; elements that are not possible through traditional CAR genetic engineering approaches. Although the adaptor format is near-limitless, the expense of generating CAR switch molecules and the requirement of continuous or multiple infusions of adaptors could curb its clinical implementation. CAR switch molecules rely on the addition of exogenous components; infusion of elements with non-endogenous origin could prove immunogenic, and the effect on patients remains untested.

The versatile nature of modular CARs and the ability for intelligent antigen targeting makes it an attractive candidate for CAR T therapy to be accessible to a wider range of cancer indications. Combining modular CAR T technology with new approaches in allogeneic CAR T treatment or natural killer (NK) cell engineering could create an extremely versatile product and a truly “off-the-shelf” therapy. Modular CAR T technology advances precision controllable engineered T cells and offers a more sophisticated approach to an already potent anti-cancer immunotherapy.

## Figures and Tables

**Figure 1 ijms-21-07222-f001:**
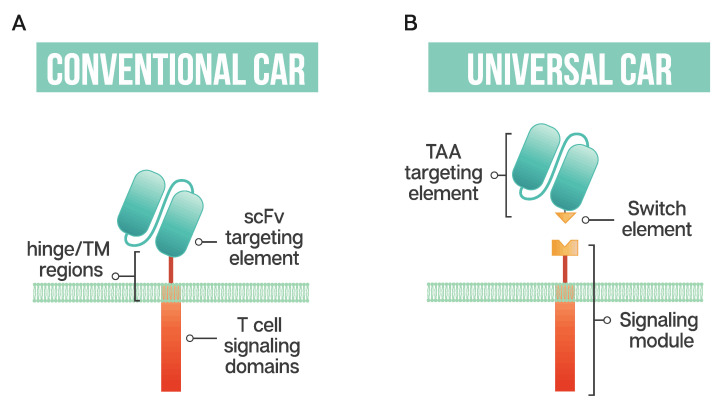
Schematic representation of a conventional CAR T cell and a universal or modular CAR T cell (**A**) Conventional CAR T cells have a single-chain antibody fragment (scFv) targeting element, expressed in tandem with signaling domains derived from the T cell receptor and costimulatory domains such as 4-1BB and CD28 connected through a transmembrane (TM) domain and a flexible spacer or hinge region; (**B**) a universal CAR T cell has a split design containing a tumor-associated antigen (TAA) targeting element, usually derived from a monoclonal antibody, a switch element and the signaling module, consisting of the T cell signaling domains and an extracellular region which interacts with the switch element.

**Figure 2 ijms-21-07222-f002:**
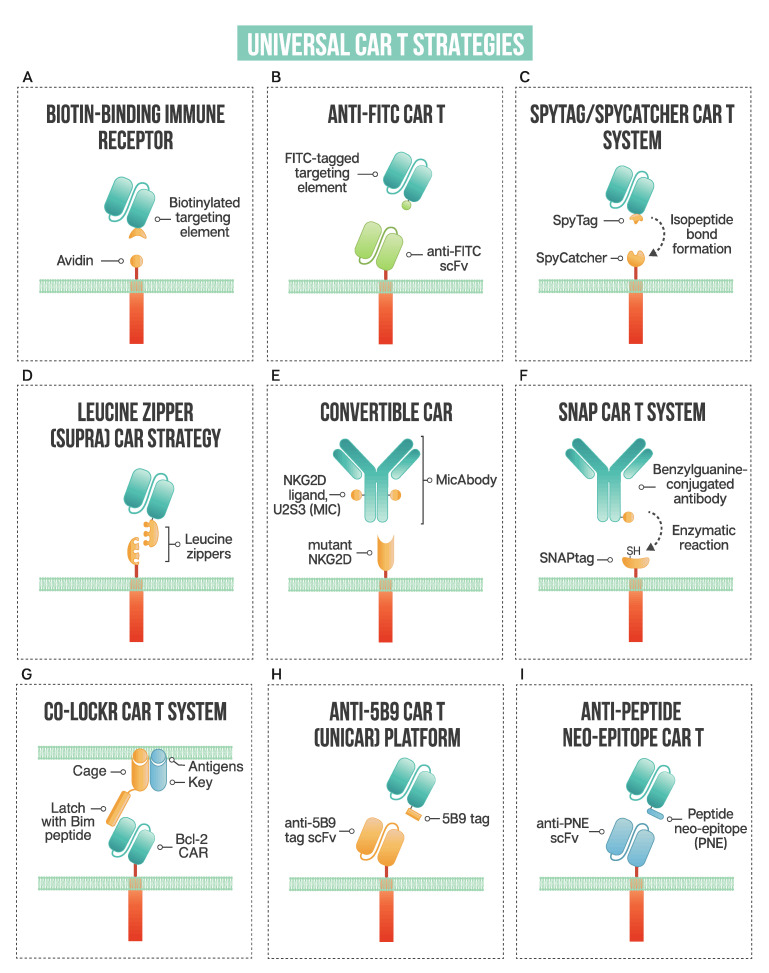
A schematic of strategies used in universal CAR T design: (**A**) biotin-binding immune receptor; (**B**) anti-FITC CAR T; (**C**) the SpyTag/SpyCatcher CAR T system; (**D**) leucine zipper or SUPRA CAR T; (**E**) convertibleCAR or modified NKG2D CAR T; (**F**) SNAP CAR T enzymatic CAR labeling system; (**G**) the co-localization-dependent protein switch (Co-LOCKR) CAR T system; (**H**) UniCAR or anti-5B9 peptide CAR platform; (**I**) anti-peptide neo-epitope (PNE) CAR T.

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
