# Peer review of "Modular Chimeric Antigen Receptor Systems for Universal CAR T Cell Retargeting"

_ijms, 2020, doi:10.3390/ijms21197222_

Round 1

Reviewer 1 Report

General:

Tremendous efforts are put in recent years into turning CAR therapy modular. Strategies explored along this route combine the infusion of T or NK cells modified to express a ‘universal’ CAR directed at a cleverly designed epitope with the in-vivo administration of an antigen-specific switch molecule (usually Abs, Ab fragments or peptides), engrafted with the CAR target epitope. Generally, this approach allows the use of a single epitope-specific CAR to target tumor or virally infected cells decorated with any antigen-binding moiety as long as it possesses the CAR epitope, so that the same CAR can be employed for numerous clinical uses. Apart from the simplicity and universality of use offered by this approach, it allows the controllable titration of timing and magnitude of T or NK cell reactivity. This includes the complete cessation of reactivity in case adverse effects are manifested, achieved by withdrawal of the switch molecule, which, in contrast to the use of suicide genes, leaves the therapeutic cells unaffected, ready for renewal of treatment.

This review initially refers to the challenges facing ‘conventional’ CAR-T cell therapy and continues with a comprehensive description of the principles underlying this approach. It then presents each strategy in detail, elaborates on the molecular design and mechanism of action (accompanied by clear sketches), refers to experimental evidence for activity and discusses anticipated advantages along with potential pitfalls.

This is an excellent review, which fits well with the scope of IJMS.

Major comment:

I highly recommend that the authors add a section describing the new switch design presented in a Science paper published online last month: Lajoie MJ et al.: Designed protein logic to target cells with precise combinations of surface antigens. Science, 2020. This paper describes a novel modular CAR system with a powerful switch, which can be exploited to implement AND, OR, and NOT logic gates. Although this Science paper may have been published after submission of the current manuscript, adding this powerful strategy to those already reviewed will significantly contribute to the strength of this review, further highlighting the versatility and potential clinical utility of modular CARs.

Minor comments:

  1. Line 130: “The mSA2 CAR T cells showed potent effector functions, however, its potential…” please check.
  2. Line148: “Additionally, the use of anti-FITC CAR T cells were shown” – should be “was shown”.
  3. Line 157: “to lung epithelial” should be epithelium.
  4. Line 243: “In both in vitro and in vivo the efficacy…” please check.

Author Response

As requested by Reviewer 1, a section covering the universal CAR system outlined in Lajoie MJ et al.: Designed protein logic to target cells with precise combinations of surface antigens. Science, 2020 was added. As well, an addition to Figure 2 was completed to illustrate this system. Additional changes to correct grammar on Line 148 and Line 157 were made.

Reviewer 2 Report

In this review Dr Sutherland and co-authors review, summarize and comment on the evidence and new opportunities offered by modular CAR strategies, highlighting advantages in overcoming current limitations with fixed-antigen CAR-T lymphocytes.

The manuscript is informative, well written and balanced, offering important insights for scientists and clinicians working in the field.

Minor comments:

In the conclusion section, when new possible perspective are discussed it could be helpful to mention and comment on the possible CAR engineering of lymphocytes different from “conventional T” cells.  For instance potential applications with NK, γδ or CIK lymphocytes might combine the versatility of modular CARs with their intrinsic (CAR-independent) antitumor properties.

Author Response

As for Reviewer 2’s comments on mentioning other non-traditional CAR systems, addition of NK cell therapy was added in line 508, however, as this is a CAR T review we did not cover γδ or CIK therapies as they are outside of the scope of this article.